# Accounting for albedo change to identify climate-positive tree cover restoration

Natalia Hasler[1], Christopher A. Williams ●[2] ✉, Vanessa Carrasco Denney[3], Peter W. Ellis ●[4], Surendra Shrestha[2], Drew E. Terasaki Hart ●[3,5], Nicholas H. Wolff ●[6], Samantha Yeo ●[3], Thomas W. Crowther ●[7], Leland K. Werden[7] & Susan C. Cook-Patton ●[3] ✉

Restoring tree cover changes albedo, which is the fraction of sunlight reflected from the Earth's surface. In most locations, these changes in albedo offset or even negate the carbon removal benefits with the latter leading to global warming. Previous efforts to quantify the global climate mitigation benefit of restoring tree cover have not accounted robustly for albedo given a lack of spatially explicit data. Here we produce maps that show that carbon-only estimates may be up to 81% too high. While dryland and boreal settings have especially severe albedo offsets, it is possible to find places that provide net-positive climate mitigation benefits in all biomes. We further find that on-the-ground projects are concentrated in these more climate-positive locations, but that the majority still face at least a 20% albedo offset. Thus, strategically deploying restoration of tree cover for maximum climate benefit requires accounting for albedo change and we provide the tools to do so.

Restoring tree cover to places that would naturally support trees is a prominent strategy for removing carbon from the atmosphere and tackling the climate crisis[1–3]. However, the net climate impact of restoring tree cover depends on more than carbon sequestration; it also alters albedo, which is the fraction of solar radiation reflected from the land surface back to the atmosphere. Because tree cover often absorbs more solar radiation than other land covers, this can lead to local[4–6] and global warming[6–14]. In some locations, global warming from albedo change can partially or even completely countervail the cooling benefit of increased carbon storage in trees[8–10,13–15].

The climate warming response to changes in surface albedo can be directly compared to changes in carbon storage by expressing them in the same unit. That unit, carbon dioxide equivalents ($CO_2e$) is calculated by first converting changes in surface albedo to top-of-atmosphere radiative forcing (TOA RF) using the radiative kernel technique[16–22]. A radiative kernel quantifies the change in outgoing radiation flux at the top of the atmosphere (radiative forcing) in

response to a change in a climate system state variable such as surface albedo. The TOA RF from albedo change is then converted to $CO_2e$ by finding the equivalent $CO_2$ pulse yielding the same TOA RF, based on the average fraction of $CO_2$ removed from the atmosphere by all global carbon sinks after 100 years[22,23]. This accounts for the time decay of $CO_2$ in the atmosphere with an impulse response function[24] describing ocean and land $CO_2$ exchange with the atmosphere.

Conversion to the same unit makes it possible to calculate "albedo offset" as the percentage of cooling from carbon storage that is offset by warming from albedo change. A 50% offset indicates that a decrease in albedo halves the net climate benefit. A greater than 100% offset indicates that a decrease in albedo entirely overwhelms carbon storage, producing a net climate-negative outcome. While restoring tree cover can impact other global-scale factors that influence climate (e.g., changes in clouds, evaporation, sensible heat flux, and other factors can alter earth's top-of-atmosphere longwave and shortwave radiation fluxes as well as surface temperatures in complex ways), quantifying

[1]George Perkins Marsh Institute, Clark University, Worcester, MA, USA. [2]Graduate School of Geography, Clark University, Worcester, MA, USA. [3]Tackle Climate Change Team, The Nature Conservancy, Arlington, VA, USA. [4]Tackle Climate Change Team, The Nature Conservancy, Portland, ME, USA. [5]CSIRO Environment, Brisbane, QLD, Australia. [6]The Nature Conservancy, Brunswick, ME, USA. [7]ETH Zurich, Zurich, Switzerland. ✉e-mail: cwilliams@clarku.edu; susan.cook-patton@tnc.org

their impact on global climate is not yet tractable[25]. Furthermore, changes in surface albedo have been shown to dominate other factors in at least some locations[11–13,26].

Although previous work has emphasized the importance of accounting for albedo change[10,15,27], albedo is either omitted from—or only coarsely modeled in—most assessments of the mitigation potential of restoring tree cover[2,3,27,28]. Prior efforts have used latitudes or biome boundaries to eliminate potentially disadvantageous areas[2], or have applied uniform and arbitrary deductions to carbon accumulation in places where large albedo changes are expected[28]. Some recent studies, using more sophisticated spatial methods, find that albedo change substantially offsets the climate benefit of restoring tree cover in Canada[29] and of afforestation in global drylands[30]. Albedo offsets are generally expected to be highest in locations with lots of sunlight, with consistent snow cover or other highly reflective land surfaces[25,30], as well as in places where trees have slow rates of carbon accumulation[30]. However, changes in surface albedo with land cover change and subsequent climate effects, can vary substantially at local scales (e.g., ref. [7]), Thus, spatially refined maps of these effects are needed to more fully characterize the climate implications of restoring tree cover across the landscape.

Here we present a series of global maps estimating the albedo-driven climate forcing resulting from restoration of tree cover and compare it to maps estimating the climate change mitigation potential of forest restoration. We first produced a set of maps quantifying changes in TOA RF due to a transition from each of four different open land cover classes (open shrubland, grasslands, cropland, or cropland/natural vegetation mosaics) to each of six different forest classes (woody savanna, evergreen needleleaf, evergreen broadleaf, deciduous needleleaf, deciduous broadleaf, or mixed forests) (Table S1). To determine where albedo change substantially reduces (i.e., >50% offset)—or entirely offsets (i.e., >100% offset)—carbon storage, we combined these 24 change maps into a single potential albedo change map. We then combined this potential albedo change map with a published map of maximum potential carbon storage[27] to predict the net climate benefit of efforts to restore tree cover across most of the globe. Finally, we use these to refine three previously published maps that identify areas to restore tree cover[2,27,31], as well as assess the albedo offset occurring in on-the-ground projects ($N = 815,654$ pixels)[32,33].

## Results

### Mapping potential albedo change in $CO_2e$

We began by producing a global map of albedo-induced changes in TOA RF for each of the 24 possible open land to forest transitions at 0.05 degrees latitude/longitude spatial resolution. For example, the grassland-to-evergreen needleleaf map assumes grassland as the starting condition and evergreen needleleaf trees as the end condition for every land pixel outside Antarctica, regardless of current grassland cover or suitability for evergreen needleleaf forest. These maps are useful for project-level planning where the starting and desired final land cover are known (see "Data availability"). We created these "single land cover transition maps" by calculating the monthly change in blue-sky albedo in each pixel using the albedo atlas developed by Gao et al.[34], as well as average monthly snow-cover and radiation conditions, that we then combined with six radiative kernels[16–21] to estimate the TOA RF (see "Methods").

To combine the 24 possible transitions into a single map, we predicted the most likely open land starting condition and most likely forest class end condition for each pixel (Fig. S1), using neighborhood analyses of current land cover maps, stratified by ecoregion (see "Methods"). The composite map (henceforth, the "potential albedo change map") estimates the albedo change induced $CO_2e$ resulting from the most likely transition at each pixel (Fig. S2a). This map does not consider existing land cover conditions; instead, it captures how

albedo would change if each pixel transitioned from one of four open land cover classes to one of six forest classes. It covers most land areas, except Antarctica and places where our neighborhood analysis could not predict a most likely forest class (e.g., core desert areas).

We find that albedo-induced $CO_2e$ ranges from 28 to −469 Mg $CO_2e$ ha$^{-1}$ (or 8 to −128 Mg Ce ha$^{-1}$; 90th CI; Fig. S2a;). Negative values indicate climate warming (i.e., a reduction of the net climate benefit) while positive values indicate cooling (i.e., an augmentation of the climate benefit). A median value of −120 Mg $CO_2e$ ha$^{-1}$ (or −33 Mg Ce ha$^{-1}$) indicates that restoring tree cover generally causes some degree of albedo-driven warming, especially in arid regions and more northern latitudes, though moderate albedo-driven cooling is possible, primarily in some tropical locations (Fig. S2a).

### Global mapping of net climate-positive outcomes

The potential albedo change map tells only part of the story. To more fully quantify where restoring tree cover can serve as a climate solution, changes in albedo must be coupled with changes in carbon storage. We combined the potential albedo change map with a previously published map of maximum potential carbon storage in plant biomass (Fig. S2b) to identify the places where albedo change would most strongly negate the climate benefit of carbon storage (Fig. 1a; hereafter, 'potential net climate impact map'). This map predicts the end-most net climate impact over longer time periods, in carbon dioxide equivalents (hereafter "maximum $CO_2e$"). We find that maximum $CO_2e$ ranges from 803 to -454 Mg $CO_2e$ ha$^{-1}$ (or 219 to −124 Mg Ce ha$^{-1}$; 90th CI). A median value of 100 Mg $CO_2e$ ha$^{-1}$ (or 27 Mg Ce ha$^{-1}$) indicates that restoring tree cover is likely to result in net climate-positive benefits, but this median value is less than half (44%) of the median value of 220 Mg $CO_2$ ha$^{-1}$ (or 60 Mg Ce ha$^{-1}$) when considering just carbon (Fig. S2b).

Even with a large albedo-driven warming, tree cover may provide substantial climate change mitigation if carbon storage outweighs the albedo change effect. Yet, mapping albedo offset (Fig. 1b) shows similar, if inverted, spatial patterns to the absolute net climate impact (Fig. 1a), indicating that the greatest albedo offsets generally occur in places that also have lowest potential carbon storage (also see Fig. S2). A median 52% albedo offset indicates that accounting for albedo change would most commonly halve maximum carbon storage. Hereafter, we refer to a > 50% albedo offset as a 'substantial albedo offset' but selecting >50% as a threshold is somewhat arbitrary and other thresholds are possible (see "Data availability").

Contrary to previous work that suggests the greatest albedo concerns are in the boreal[2,10], we find that drylands have a greater proportion of net climate-negative areas (Fig. 1a; Fig. 2). In particular, 72% of the temperate grasslands, savannas, and shrubland biome would be climate-negative, and 83% of the biome would experience a substantial albedo offset (Fig. 2C, Table S2). Across the Mediterranean forests, woodlands, and scrub biome, 60% of the area would be climate-negative (76% would experience a substantial albedo offset, Table S2) and across the tropical and subtropical grasslands, savannas, and shrubland biome 38% would be net climate-negative (46% would experience a substantial albedo offset; Fig. 2D; Table S2). In comparison, 34% of the total area in the boreal forest biome would be net climate-negative, but 72% would experience a substantial albedo offset (Fig. 2D; Table S2). Thus, despite the lower proportion of net climate-negative areas in the boreal relative to these dryland settings, changes in albedo remain a concern across most of the boreal. At the other end of the spectrum, only 3% of the total area in the tropical and subtropical moist broadleaf forests biome is predicted to be net climate-negative (and only 6% would experience a substantial albedo offset, Fig. 2A; Table S2).

Despite these general biome-level patterns, it is important to flag that variation exists within biomes and that all biomes have at least some net climate-positive locations (Fig. 2). For example, in the

tropical and subtropical grasslands, savanna, and shrubland biome there are areas where restoring tree cover would result in net climate-negative impacts and areas where changes in albedo are of little concern (Fig. 2D). This indicates the importance of spatially refined maps for both carbon and albedo change.

## Refining areas of opportunity

Much of the area in Figs. 1 and 2 already supports tree cover or is under human land use, and only a fraction could experience restoration of tree cover. To determine where restoring tree cover can serve as an actual climate solution, we used our potential albedo offset map to refine three previously published maps that identify areas of opportunity for tree cover restoration.

The Griscom opportunity map includes 828 million hectares (Mha) that are biophysically suitable for forests, not currently under a crop or urban land use, nor in a grassy biome[2]. It is the only map of the three that considered possible albedo offsets, simply excluding boreal ecoregions. While most of the area (94%) would result in net climate-positive outcomes, we find that 18% would experience a substantial albedo offset (Fig. 3a-b; Table S2). Moreover, if forests were restored across the entire area of opportunity, maximum $CO_2e$ would shrink 20% after accounting for changes in albedo (from 318 to 254 Pg $CO_2e$, or 87 to 69 Pg Ce; Table S2).

Bastin et al.[31] estimate the potential to increase tree cover, even in locations that already have forests. Thus, to examine a more comparable map to the Griscom opportunity map, we identified the 916 Mha with <25% tree cover that could be suitable for >25% tree cover. As with the Griscom opportunity map, most of the area of opportunity (71%) would result in net climate-positive outcomes, but we find that almost half (48%) would experience a substantial albedo offset (Fig. 3c, d, Table S2). Moreover, if forests were restored across the entire area, maximum $CO_2e$ would be halved (53%) due to changes in albedo (from 214 to 101 Pg $CO_2e$, or 58 to 28 Pg Ce; Table S2). Notably, it is possible to achieve greater maximum $CO_2e$ (127 Pg $CO_2e$ or 35 Pg Ce) by restoring only half of the total area (52% or 476 Mha) rather than the entire area, by restricting activities to places that do not experience a substantial albedo offset (Table S2).

The Walker opportunity map[27] identifies 889 Mha as currently possessing low carbon storage but with the potential to support forest-level amounts of carbon. However, only half of the total area (54%) would be climate-positive and 65% would experience a substantial albedo offset (Fig. 3e, f; Table S2). If forests were restored across the entire area of opportunity, changes in albedo would reduce maximum $CO_2e$ by 81% due to the inclusion of some very climate-negative areas (from 186 to 35 Pg $CO_2e$, or 51 to 10 Pg Ce; Table S2). In contrast, targeting only areas without a substantial

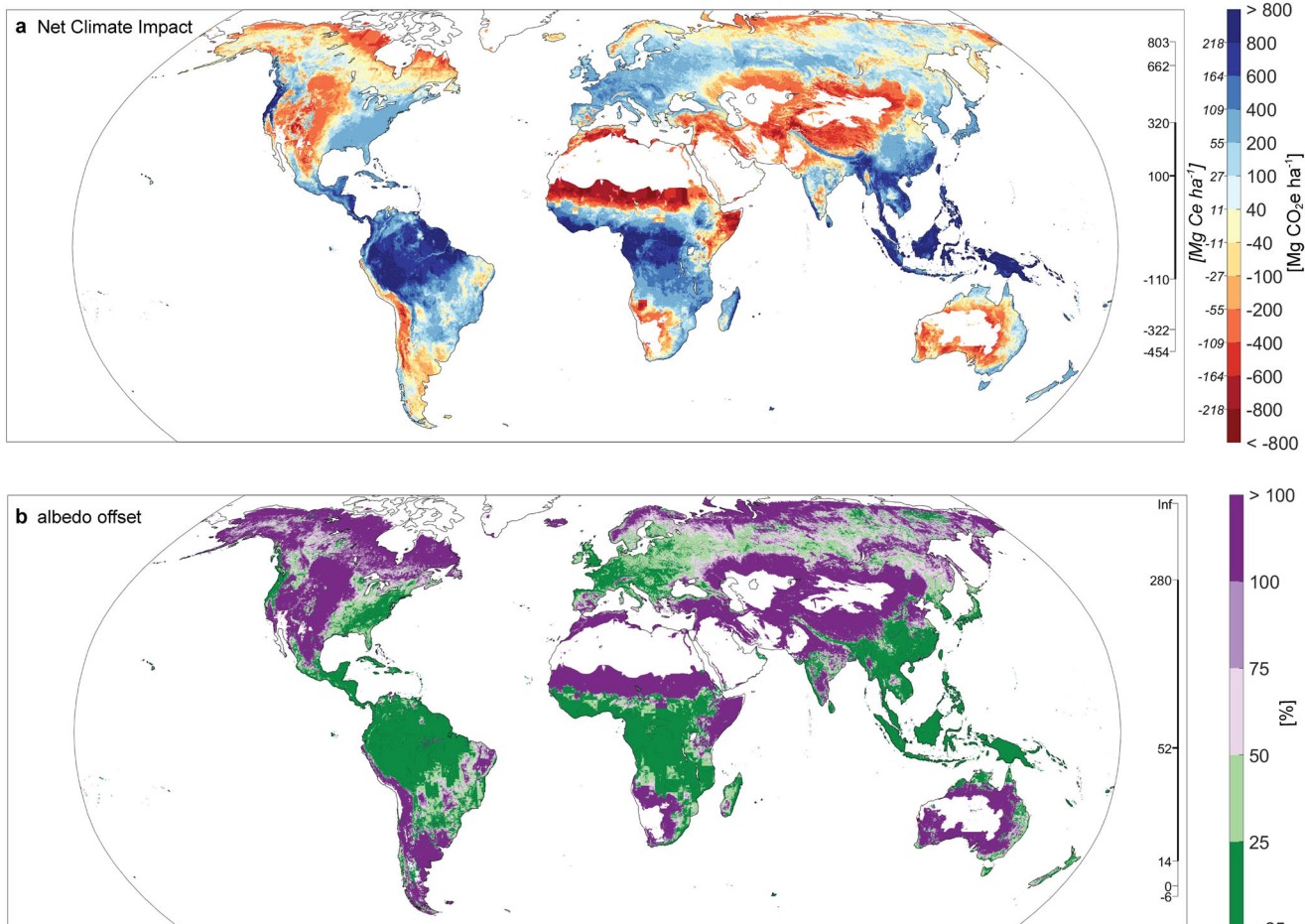

**Fig. 1 | Net climate impact (in megagrams carbon dioxide equivalents per hectare, Mg $CO_2e$ ha$^{-1}$) and albedo offset (%). a** The net climate impact accounts for both albedo change and carbon storage to estimate maximum $CO_2e$. Orange colors indicate net climate-negative locations, whereas blues indicate net climate-positive. For comparison to other studies, we also provide estimates in carbon equivalents (Mg Ce, italicized text to the left of the color ramp). **b** Albedo offset is the percent of maximum carbon storage offset by changes in albedo. Purple colors indicate locations where albedo offsets >50% of maximum carbon storage, whereas green indicates <50% albedo offset. In both maps, data are binned for display purposes and the scale bar immediately to the right of the maps indicates the 5%, 10%, 25%, 50%, 75%, 90%, and 95% land-area percentiles (top to bottom). Source data are provided as a Source data file (see "Data availability").

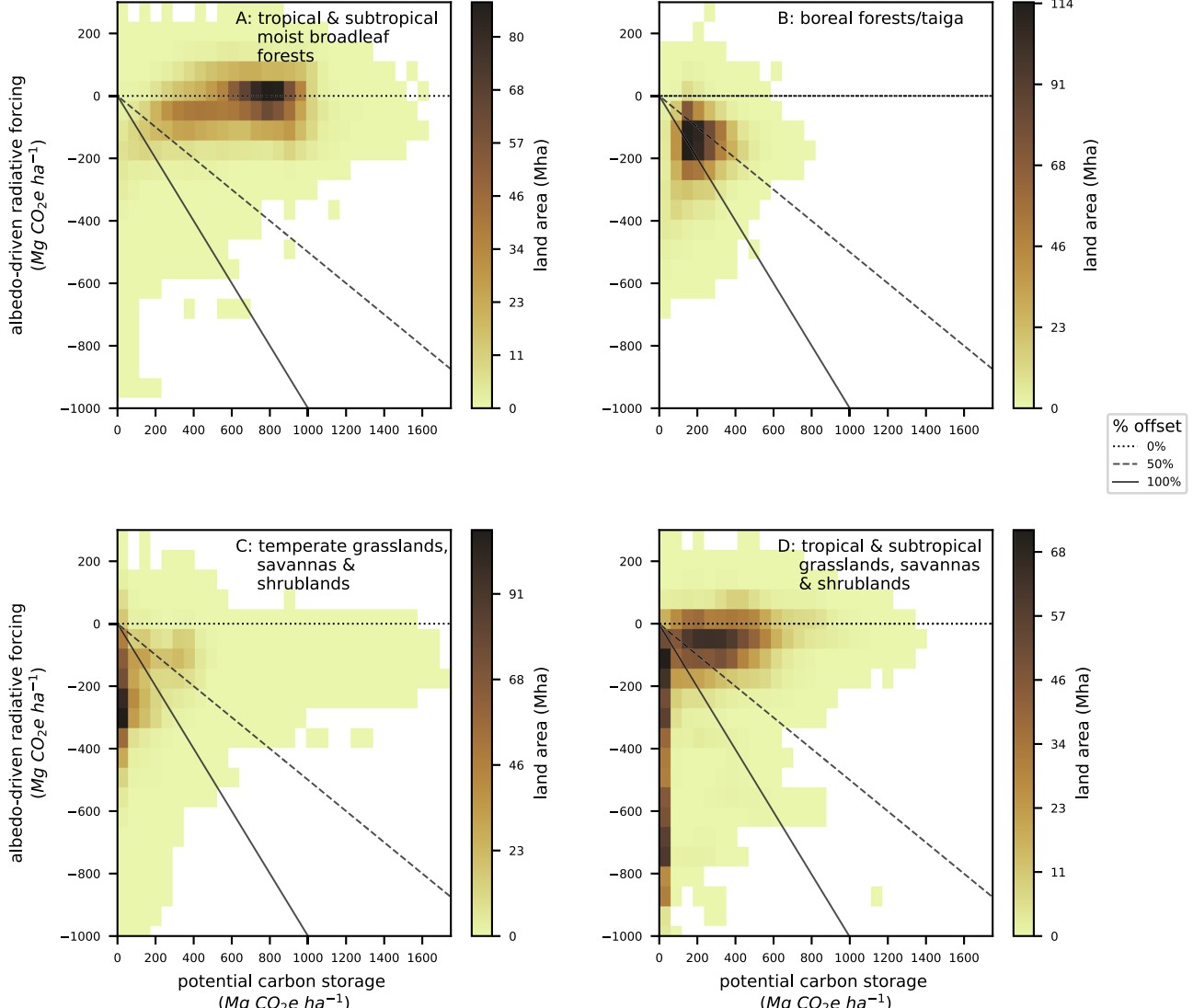

**Fig. 2 | Variation in carbon storage and albedo change across and within biomes (in megagrams of carbon dioxide equivalents per hectare, MgCO₂e ha⁻¹).** Within biomes, both potential carbon storage (x-axis) and albedo change (y-axis) can vary. These panels depict the consequences of transitioning from the most likely open to forest land cover at every pixel in that biome (irrespective of current land cover or suitability for restoring tree cover). The colors indicate the amount of area with a specific albedo change and carbon combination (see color bar). Pixels that fall along the solid diagonal line have zero net climate impact (albedo offsets carbon storage perfectly). Pixels along the dashed diagonal lines correspond to 50% albedo offset and the horizontal dashed line indicates areas with no albedo offset. We show four biomes with divergent patterns: **A** a biome where albedo change is of low concern, **B** a biome where changes in albedo offset much but not all of the climate benefit, **C** a biome where changes in albedo offset most of the climate benefit, and **D** a biome where there are both low and high albedo offsets. Source data are provided as a Source data file (see "Data availability").

albedo offset (311 Mha or 35%) rather than the entire area would achieve 2.5-fold more maximum CO₂e (90 Pg CO₂e or 25 Pg Ce; Table S2).

These opportunity maps use different methods to identify areas of opportunity and show quite divergent patterns. Our results suggest that almost a fifth, almost a half, and over two-thirds of the area of opportunity identified in Griscom et al.[2], Bastin et al.[31], and Walker et al.[11] respectively, may not be suitable for restoration of tree cover as a climate solution, because they would experience a substantial albedo offset. However, they all agree that biomes with the most positive net climate outcomes include tropical and subtropical moist broadleaf forests (65 Pg CO₂e, or 18 Pg Ce, on average) and temperate broadleaf and mixed forests (30 Pg CO₂e, or 8 Pg Ce, on average) (Table S2; Fig. 3). The Walker and Bastin opportunity maps also jointly identify substantial opportunity in tropical and subtropical grasslands, savannas, and shrublands (Table S2; Fig. 3).

**Accounting for albedo change in on-the-ground projects**

These opportunity maps depict potential and not actual projects. To understand how albedo change impacts the climate outcomes of recent, planned, and ongoing activities to restore tree cover, we assembled spatial data from the Grain for Green program (a large-scale restoration project across degraded farmlands in China[33]) and Restor (a data sharing platform that allows practitioners to map their projects and facilitates global efforts to restore and conserve nature)[32]. We examined all pixels in our net climate impact map that overlapped with a project (hereafter "project pixels"). Notably only 45% of these overlap with at least one of the opportunity maps (Fig. 4). This indicates a mismatch between where global maps project potential for restoration of tree cover versus where actual projects occur. We find that 84% of the 815,654 project pixels occurred in net climate-positive locations (Figs. S3–S4) and 29% had a substantial albedo offset (i.e., a > 50% albedo offset; Fig. 4; Table S3). Moreover,

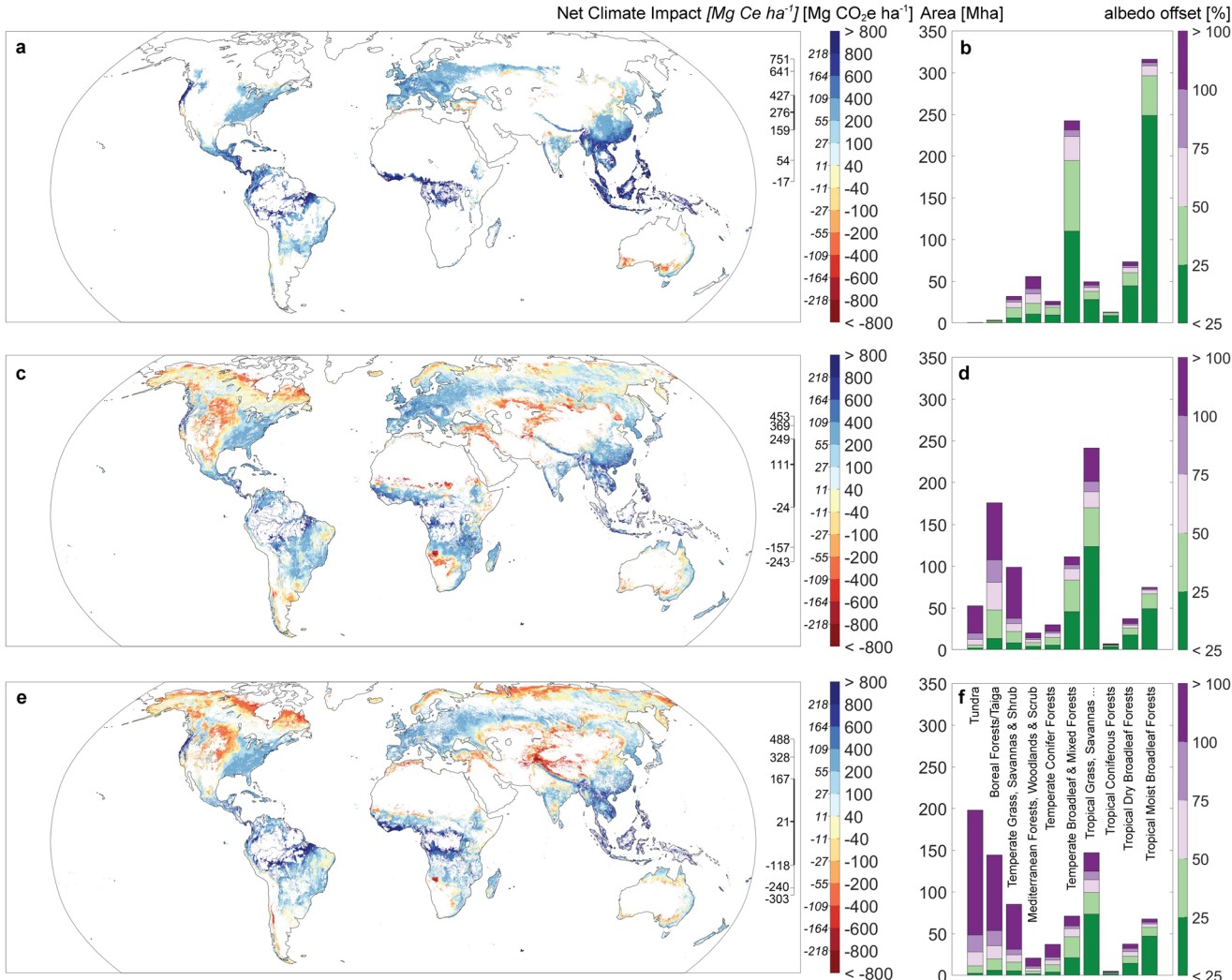

**Fig. 3 | Net climate impact (in megagrams of carbon dioxide equivalents, Mg CO₂e) and albedo offset (%) in published opportunity maps. a**, **b** Griscom et al.[2], **c**, **d** Bastin et al.[13], and **e**, **f** Walker et al.[11]. The scale bar to the immediate right of the maps indicates the 5%, 10%, 25%, 50%, 75%, 90%, and 95% land-area percentiles (top to bottom). For comparison to other studies, we also provide estimates in carbon equivalents (Mg Ce, italicized text to the left of the color ramp). Source data for maps are provided as a Source data file (see "Data availability"). Bar charts provide biome-level summaries. There is some additional opportunity that occurred in non-forest biomes (montane grasslands, flooded grasslands, mangroves, and deserts, see Table S2).

the majority (66%) had at least a 20% albedo offset (Fig. 4; Table S3), highlighting the need to account for changes in albedo if climate impacts are a project goal.

## Uncertainty
Multiple radiative kernels have been developed by several climate modeling teams to support computationally efficient assessment of feedbacks in the climate system[16–21]. We used the six different radiative kernels that are currently available (five from different global climate models[16,18–21] and one from a radiation budget model[17] to estimate a range of outcomes for albedo change. We found that uncertainties around the net climate impact are small overall—generally in the range of +/−15% of the median estimate or less, except in the high latitude areas (tundra and boreal forest biomes) (Table S2). Moreover, when we account for the full range across all radiative kernels, very few pixels transition above or below the substantial albedo offset category (i.e., >50% albedo offset; Fig. S5). Except for a few locations in transitional zones (9% of land area), most pixels were consistent across kernels in having either a greater than 50% albedo offset (36% of area) or a less than 50% albedo offset (35% of area; Fig. S5).

Our results are also sensitive to the carbon dataset used, and the one used here may be too high in places[35]. As a sensitivity test, we therefore used the 85% percentile of current carbon storage[33] to truncate the highest values in the maximum potential carbon storage layer[27] (Fig. S6). This decreased the net climate impact globally by 14% from a median of 100 Mg CO₂e ha⁻¹ to 86 Mg CO₂e ha⁻¹ (or 27 Mg Ce ha⁻¹ to 23 Mg Ce ha⁻¹). Despite this decrease, neither the extent of net climate-negative areas nor areas with substantial albedo offsets changed substantially.

## Discussion
Changes in albedo are a commonly cited concern for climate change mitigation initiatives in the boreal[2,10,25]. While we find substantial albedo offsets in the boreal zone, dryland settings showed a greater proportion of net climate-negative areas (Figs. 1 and 2). This aligns with a recent analysis in global drylands that also predicted limited climate change mitigation from afforestation after accounting for albedo[30]. It also further underscores the inadvisability of afforestation of native grasslands, which can negatively impact biodiversity[36], and result in high tree mortality and project failure[37]. However, all biomes had at least some climate-positive locations (Table S2), highlighting the

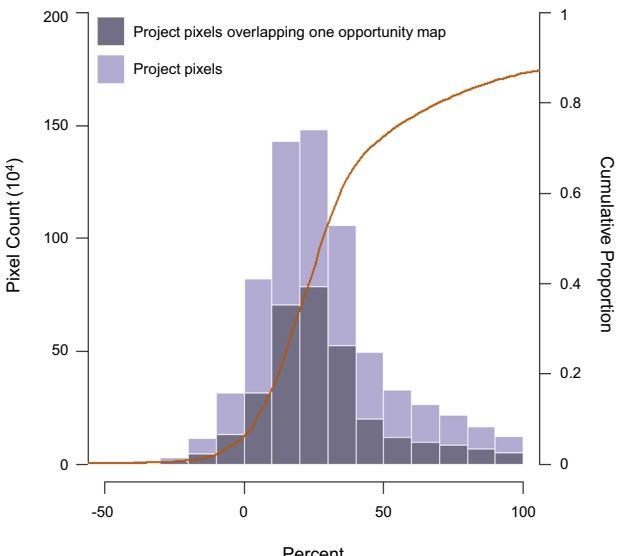

**Fig. 4 | Distribution of albedo offset for on-the-ground projects.** These project pixels represent past, on-going, and planned projects that are part of the Grain for Green Program[33] or uploaded onto Restor[35]. The overall height of the bar shows how often the project pixels (Fig. S4) overlap with different albedo offset bins (left y-axis). The darker gray shading indicates the number of pixels that overlap with at least one of the opportunity maps[2,27,31]. The red line indicates the cumulative proportion of projects (right y-axis). This figure does not include *NA* values (Table S3).

importance of spatially explicit albedo accounting. Overall, changes in albedo will likely offset some percentage of carbon storage across most of the pixels we mapped. This includes pixels where on-the-ground projects are occurring, though projects that seek to restore tree cover are concentrated in net climate-positive locations (Fig. S4). Compared to on-the-ground projects, previously published opportunity maps were even more negatively impacted, with 20 to 81% reductions in maximum $CO_2e$.

The three previously published maps identify quite strikingly different areas of opportunity (Fig. 3), highlighting the need for reconciliation and improvement. Yet all experienced a reduction after accounting for albedo change. In the Walker opportunity map, with its heavier concentration of opportunity in tundra, tropical savanna, and boreal biomes, accounting for albedo reduced maximum $CO_2e$ by 81% and area of opportunity by 65% (311 Mha with <50% albedo offset relative to total area). In contrast, the Griscom opportunity map was the least impacted by accounting for albedo change. It was the only one of the three that deliberately sought to eliminate areas due to albedo concerns, but also likely misses climate-positive opportunities to restore tree cover due to an overly restrictive exclusion of the entire boreal[2]. The map also excludes grassy biomes for biodiversity reasons, which inadvertently avoids areas with large decreases in albedo from restoring tree cover.

Ultimately, only a fraction of the maximum $CO_2e$ can be achieved given competing land uses (e.g., agricultural production) and the priorities of local communities[38,39], but it is worth noting that higher climate mitigation can be achieved by restoring less area, if activities avoid net climate-negative areas. For example, restoring approximately a third of the Walker opportunity map offers 2.5 times more maximum $CO_2e$ than restoring the whole area. These spatially explicit albedo change maps can help to concentrate limited funds towards restoration opportunities with the highest net climate impacts.

In contrast to opportunity maps, actual projects were generally concentrated in net climate-positive locations (84% of project pixels), with 29% experiencing a substantial albedo offset. This suggests that on-the-ground activities are already being targeted towards locations

with less severe changes in albedo. However, two-thirds will experience an albedo offset of 20% or more, which indicates that if climate change mitigation is a goal, projects need to account for albedo change, especially in situations where restoring tree cover is used to generate offsets.

Finally, although we focus on climate change mitigation in this paper, there are many additional benefits from restoring tree cover, such as creation of habitat[40–42], improved livelihoods[43], enhanced hydrological benefits[44]. Moreover, removal of atmospheric $CO_2$ confers additional environmental benefit by helping to mitigate ocean acidification[45]. Our work does not aim to dictate where tree cover should be restored, but rather to help better account for net climate impacts when mitigation is a goal.

We accounted for uncertainty in our estimates by taking advantage of the variation across radiative kernels[16–21]. Other potential sources of uncertainty, for which we could not account include uncertainty in the land cover class designation (in the MODIS land cover product, as well as our designation of likely open and forest classes), and in the MODIS snow cover and the black- and white-sky radiation archives. Our net climate estimates also depend on the data source used to map potential carbon storage (Fig. S2b). Although we show the results for a modified carbon layer (Fig. S6), more detailed estimates of potential carbon accumulation at a given location are likely needed for project-level estimates of net climate impacts.

Moreover, we compared maximum potential carbon storage[27] with maximum albedo change, which ignores the time frame over which both unfold. Maximum $CO_2e$ is a threshold value that would be achieved over longer time horizons (i.e., decades to centuries depending on forest growth rates[46]), with both being low during the early stages of stand development as trees grow and the canopy expands[29]. We use maximum $CO_2e$ because these temporal dynamics of albedo changes are not well described. However, modeling different scenarios of carbon and albedo change through time (see "Methods" and Fig. S7) shows that many areas will remain net climate-positive, or net climate-negative, regardless of the time horizon. Yet, in other areas, particularly where the albedo offset is intermediate, it is possible for the net climate impact to range from positive to negative depending on the time horizon (Fig. S7). Given that restoring tree cover is often highlighted as a readily deployable and scalable solution in the near-term[3,47], additional work is urgently needed to determine the net climate impact over the coming decades.

In addition to temporally explicit estimates, there is likely a need for even more spatially refined estimates. We produced a map nominally at 500-m resolution (though we sampled more widely to determine the most likely open and forest land class, see "Methods"). This represents a substantial advance over previous work that relied on latitudinal bands or ecoregional boundaries. However, there is likely much finer variation in albedo due to factors such as topography and aspect. Local topography influences illumination angles and thus the intensity of solar radiation. Topography also influences vegetation distributions and associated biophysical properties. Both can meaningfully affect surface albedo. While moderate to high-resolution datasets could capture this local variability, it is only weakly captured in available spaceborne albedo datasets that have global coverage. Thus, an important future direction is higher-resolution analysis in regions with strong albedo changes, and/or regions where topography influences plant species distributions (e.g., where evergreen needleleaf are appropriate only on poleward-facing slopes).

Future climate conditions may also alter the climate response to tree cover restoration in ways not captured in this study. At high latitudes and altitudes, climate change is expected to reduce persistent snow cover[48] and temperature limitations on growth[49], both of which would reduce the albedo offset from restoring tree cover. However, the negative climate impacts in drylands will persist, or even grow if the extent of dryland biomes increases and tree growth is further

diminished as the climate warms[50]. Beyond climate change, future timber demands will likely alter the extent, management (e.g., rotation lengths), and types of forest (e.g., when an evergreen timber plantation replaces a deciduous native forest)[51]. All of these changes will impact both carbon storage within forests and albedo[52].

Although accounting for both albedo change and carbon storage is an improvement over exclusively quantifying carbon, the net climate impact of restoring tree cover depends on multiple additional factors that we did not capture here. The currently available radiative kernels that we used in this study represent instantaneous rather than effective radiative forcing and omit some of the atmospheric adjustments that could be expected to reduce the albedo-related forcing to some degree[53,54]. Trees can emit methane, which would reduce the climate benefit[55]. Restoring tree cover can impact surface temperatures and cloud formation through changes in surface evaporation, surface roughness, or the release of volatile organic compounds, all of which can influence global climate in complex ways that are only weakly characterized at present[25]. Restoring tree cover can also help to restore precipitation patterns in regions with high rates of precipitation recycling, potentially helping to protect downwind forests from drought-driven mortality[56,57], which would further augment the net climate benefit. Additional consideration of the durability of tree cover is also warranted to avoid disturbance prone areas, especially those seeing increasing trends of wildfire, insect outbreaks, or other tree-killing events[58]. While we did not account for these factors in this analysis, estimating the full net climate impact will ultimately require much fuller accounting across all such factors.

Restoring tree cover is not a panacea for climate change. It is also critical to reduce fossil fuel emissions and protect intact ecosystems. However, restoring tree cover remains a promising natural climate solution for removing carbon dioxide from the atmosphere if it is located in climate-positive locations. Our work shows the need to account for albedo change when restoring tree cover for climate change mitigation and provides the tools to do so in a robust and spatially explicit way.

## Methods

### Mapping albedo change for 24 land cover transitions

The methods used to estimate changes in albedo for 24 different open land to forest land cover transitions are similar to those employed by other studies[7,8,29,59]. We first identified land as all pixels with ≥1% of their area covered by any land cover type other than water (according to the MODIS/Terra+Aqua Land Cover Type version 6, MCD12C1, 2016) and also excluded any land south of 60S (i.e., Antarctica). Then for every land pixel in the MODIS Climate Modeling Grid (CMG), we calculated the albedo-induced changes in top-of-atmosphere radiative forcing (TOA RF) of an instantaneous conversion from each of four open land classes (open shrublands, grasslands, croplands, and cropland/natural vegetation mosaics) to each of six forest or savanna classes (evergreen needleleaf forests, evergreen broadleaf forests, deciduous needleleaf forests, deciduous broadleaf forests, mixed forests, and woody savanna). See Table S1 for descriptions of the land cover classes, acronyms, and codes.

For these first calculations, we estimated each transition across all global lands regardless of whether the initial and final land covers are plausible. To calculate changes in surface albedo, we used the albedo atlas developed by Gao et al.[34]. This atlas provides unique black- and white-sky albedo values for each combination of month, MODIS-CMG pixel and International Geosphere Biosphere Programme (IGBP) land cover class under snow-covered and snow-free conditions. For monthly snow conditions, we used all available monthly data from MODIS/Terra Snow Cover Daily L3 Global 0.05Deg CMG (Version 6.1) to determine average monthly conditions across the two decades of available data (March 2000 to August 2021). For monthly black and white sky conditions, we used a product from the National Center for

Atmospheric Research (NCAR) National Centers for Environmental Prediction (NCEP)[60,61]. This product provides a diffuse and beam incoming surface solar radiation (visible and near-infrared) reanalysis Gaussian grid (T62 with 94 × 192 points) from 1981 to 2010 and was resampled to this study's 0.05Deg grid with nearest neighbors. We combined visible and near-infrared components in the calculation of blue-sky albedo (Eq. 1).

Change in surface albedo is the difference between blue-sky albedo conditions of the initial land cover (lc1) and converted land cover (lc2), as a per area unit. It is a weighted combination over direct (black sky, $r = 0$) and diffuse (white sky, $r = 1$) illumination conditions (fractions, $f_{r,m}$) and over snow-covered and snow-free conditions (fractions, $f_{s,m}$) specific to each grid-cell (x,y) and month (m):

$$\Delta\alpha_m(x,y) = \sum_{r=0}^{1}\sum_{s=0}^{1} f_{r,m}(x,y) \times f_{s,m}(x,y)(\alpha_{lc2,r,s,m}(x,y) - \alpha_{lc1,r,s,m}(x,y))$$
(1)

To estimate TOA RF due to surface albedo change, we calculate monthly TOA RF within areas locally undergoing a land conversion. We used the grid-cell specific albedo radiative kernels ($K_m$) generated by six different models, resampled to match the MODIS-CMG grid, with nearest neighbors. These models include five different global climate models, specifically CAM3[19], CAM5[18], ECHAM6[16], HadGEM2[20], HadGEM3[21], and one radiation budget model, CACKv1.0[17].

$$RF_{\alpha,m}(x,y) = K_{\alpha,m}(x,y) \times \Delta\alpha_m(x,y)$$
(2)

Because we had multiple radiative kernels, we used the pixel-wise median value across these kernels but provide minimum and maximum values to capture the range of possible values (see "Data availability"). Finally, we estimated an annual TOA RF, based on a simple average across months and converted TOA RF to the global scale by multiplying by the grid cell area ($A_{gridcell}$) ratio to global earth surface area ($A_{global}$).

$$RF_{\alpha,global}(x,y) = RF_{\alpha}(x,y)\frac{A_{gridcell}(x,y)}{A_{global}}$$
(3)

### Converting radiative forcing to carbon dioxide equivalents

To facilitate comparisons between carbon storage and albedo change, we converted radiative forcing into $CO_2e$. To do so, we adopt the global annual mean radiative forcing caused by carbon emissions per square meter of global surface area from the IPCC[62], describing a perturbation to Earth's top-of-atmosphere radiation budget imposed by a change in global atmospheric $CO_2$ concentration as:

$$RF_{CO_2} = 5.35\ln\left(\frac{C_a(t)}{C_a(t_0)}\right)$$
(4)

where $C_a(t)$ is the mass, in Pg, of C in the atmosphere equal to 2.13 times the $CO_2$ concentration in ppm at time $t$, and $C_a(t_0)$ is the pre-industrial mass of atmospheric $CO_2$ taken from a concentration of 270 ppm.

We assume that the land cover albedo-induced changes in radiative forcing ($RF_{CO_2}$) would be equivalent to the radiative forcing of a pulse of $CO_2$ on top of a current $CO_2$ concentration, with current global mean concentration of 400 ppm ($C_{a,2020}$) and a new global mean concentration including the $CO_2$ emissions (or uptake) from restoring tree cover ($C_{a,new}$), as:

$$RF_{\alpha,global}(x,y) = \Delta RF_{CO_2} = 5.35\ln\left(\frac{C_{a,new}}{C_{a,preindustrial}}\right) - 5.35\ln\left(\frac{C_{a,2020}}{C_{a,preindustrial}}\right)$$
(5)

Solving and simplifying Eq. 5 for $C_{new}$ results in

$$C_{a,new} = C_{a,2020} e^{\frac{\Delta RF_{CO_2}}{5.35}} \qquad (6)$$

The $CO_2$ emissions (or uptake) pulse from restoring tree cover ($\Delta C_{CO_2e}$) is the difference between this new atmospheric concentration and the 2020 baseline.

$$\Delta C_{CO_2e} = C_{a,new} - C_{a,2020} \qquad (7)$$

We divide the corresponding $CO_2$ emissions (or uptake) by grid cell area to obtain the equivalent carbon mass flux per unit area. To provide those units in tons of carbon dioxide equivalent ($CO_2e$), we multiplied by the mass ratio of $CO_2$/C (44/12) yielding $CO_2e$ in Mg $CO_2e$ ha$^{-1}$.

Lastly, we normalize the $CO_2e$ from albedo change to account for the time decay of $CO_2$ in the atmosphere as described by an impulse response function[24] for ocean and land $CO_2$ exchange with the atmosphere, thus dividing the output of Eq. 7 by the average $CO_2$ fraction remaining in the atmosphere after a representative 100 years following a unit pulse emission of $CO_2$

## Mapping potential albedo change

To create a single, globally comprehensive map of potential albedo change from restoring tree cover (Fig. S2a), we first had to assign the most likely open class to all pixels that did not currently have an open land cover class and vice versa a most likely forest class to all pixels that currently did not have woody savanna or forest land cover class (Fig. S1). We did this by using a neighborhood analysis to determine the most common forest or open class within an ecoregion[63]. We followed the same methods to create the most likely open and forest class maps.

To conduct this neighborhood analysis, we created a series of nested grids of 0.01, 0.025, 0.05, 0.1, 0.25, 0.5, 1, 2.5, 5, and 10-degree resolutions, where we generally assigned each grid-pixel the value of the open (or forest) land class with the most area (i.e., the dominant class) within the ecoregion that it overlapped[63]. That class had to cover at least 1% of the ecoregion area within the grid cell. We sampled across these increasingly larger neighborhoods to determine the most likely open (or forest) class.

Specifically, we used the MODIS Terra + Aqua Land Cover Type Yearly L3 Global 500 m SIN Grid product of 2010 and 2001 (Product MCD12Q1[64]) and IGBP classification as the base for current land cover[65,66]. For open land, we used IGBP classes open shrubland (7), grasslands (10), croplands (12), and cropland/natural vegetation mosaics (14). For forests, we used evergreen needleleaf forests (1), evergreen broadleaf forests (2), deciduous needleleaf forests (3), deciduous broadleaf forests (4), mixed forests (5), and woody savannas (8) (Table S1). Although only some savanna areas are appropriate for substantial tree cover, we included woody savannas as a potential end state since those ecosystems are widespread and often inappropriately targeted by tree planting projects[36], so it is important to consider whether and how albedo change outweighs carbon storage in these locations. We also refer to woody savannas as a forest class in the main text for simplicity but note that woody savannas are generally not suitable for dense forest cover.

While we do not consider (non-woody) savannas as suitable for tree planting, we did combine savannas (9) and woody savannas (8) MODIS pixels and called them all woody savanna to determine the dominant class for the grid-pixels. We did this because in some overall savanna locations there were more MODIS forest class pixels (e.g., deciduous broadleaf forests) than woody savanna pixels. If we ignored savanna pixels entirely, those locations would have been inappropriately assigned a forest class. Moreover, labeling savannas as woody savannas is a conservative choice because the latter has a larger albedo offset than the former.

Finally, MODIS will classify disturbed or patchy forests as woody savannas even in places that would naturally support forest. Therefore, in our most likely forest map, we conservatively assigned a forest class instead of woody savanna in all ecoregions labeled forest, taiga, mangrove, várzea, pantanos, yungas, piney, and pine barren[63]. This was a conservative choice because transitioning from open land to woody savannas generally has a lower albedo offset than the forest classes. Thus, by assigning a class with a larger albedo offset, these locations are less likely to show up as net climate-positive.

Finally, because some locations had very few open (or forest) MODIS pixels from which to determine the most likely open or forest class for a given grid-pixel, we also determined the dominant open (or forest) class for each ecoregion, as well as for each Koeppen-Geiger climate zone[66] and for 25 regions in the world within biomes[67], and finally over the entire biome worldwide. We further set a minimum threshold of at least 1% of total area or 20 pixels (whichever is largest) for ecoregions and 1% area or 100 pixels for biomes. If that minimum threshold was not met, we assigned no land class to that grid-pixel, which is why there are no data for core desert areas.

To produce the final most likely maps (neighborhood analysis) we looped through the different layers in the following order until an open (or woody savanna/forest) land class was found: current MODIS (2001 and 2010 land cover), the nested grids (from finer to coarser resolutions), ecoregion, climate-biome at the region level and finally biome worldwide. In most cases, over 95% of the most likely open land class and over 75% of the forest class assigned was found within the 10° grid (see Fig. S8). Notable exceptions are deserts, montane grasslands, tundra or temperate grasslands where trees are unlikely to grow and thus have low restoration potential regardless of final forest cover (see "Data availability").

## Identifying net climate-positive areas

Once we had determined a possible open land to forest class transition for each pixel, we sampled from the kernel-median albedo change map for that transition to assemble a composite potential albedo change map (Fig. 1a). To determine locations that will result in net climate-positive locations, we incorporated a map of potential carbon storage[27]. This map provides maximum potential carbon storage in aboveground and belowground biomass, as well as soil. Because soil accumulation can take a long time to recover[68] and may not increase after restoring tree cover[1,69], we conservatively used only above- and below-ground biomass here. This potential carbon storage layer combined with the albedo $CO_2e$ layer constitute what we call the net climate impact map, representing the maximum $CO_2e$ of tree cover restoration when forest carbon accumulation and albedo change both reach their maximum.

## Characterizing likely temporal dynamics

We also modeled scenarios of how net climate impact responds to albedo and carbon changes that unfold through time to examine how conclusions might change. We began with an assumption that maximum albedo change is likely to occur before maximum carbon storage, because forests continue to accumulate carbon even after reaching full canopy extent. The relative rate of change between albedo and carbon likely varies across the globe given different rates of forest growth[1]. However, global spatial layers predicting how albedo and carbon change through time with forest development are not yet available. Thus, we model twelve hypothetical scenarios, varying the rate of carbon accumulation (fast or slow, see Fig. S9), the degree to which albedo reaches its maximum before carbon, and the magnitude of albedo offset to carbon accumulation. We assumed a carbon removal of 20 kg C m$^{-2}$ (200 Mg C ha$^{-1}$) but the value is arbitrary because other factors are expressed as a percent of maximum carbon removal or timing. We set the magnitude of the albedo offset at 100%, 50%, or 10%. We varied the timing of the albedo change to be twice as

fast (50% earlier) as carbon in reaching its maximum, or 1.25 times as fast (80% earlier). We modeled carbon accumulation with a Chapman Richards S-curve function, adjusting parameters to obtain a fast or slow approach to maximum carbon storage (Fig. S7). We accounted for ocean and land releases of $CO_2$ in response to each year's carbon removals from tree cover restoration using the multi-model mean impulse response function from a published synthesis of sixteen earth system models[24].

### Uncertainty analyses

To estimate variation in albedo change, we took advantage of the six different radiative kernels that are currently available[16–21]. For every pixel in our potential albedo change map, we sampled the radiative kernel with the largest as well as the radiative kernel with smallest albedo-induced changes in radiative forcing to determine a potential maximum range in values.

Finally, the results of net climate-positive areas are sensitive to the carbon map used. We also produce an alternative net climate impact map (Fig. S6) that truncates the Walker et al.[27] potential aboveground biomass map to be no higher than the 85th percentile of the values reported in the European Space Agency (ESA) Climate Change Initiative (CCI) forest aboveground biomass data product, version 3[70]. The ESA-CCI dataset is based on observations. It thus represents current biomass and reflects reductions from potential maximum biomass due to natural and anthropogenic disturbances. As a sensitivity test, we used the 85th percentile of the ESA-CCI dataset within each ecoregion (>800 globally)[63] as an upper bound for the Walker et al.[27] potential aboveground biomass map, to which we added the belowground biomass value calculated with unchanged root-to-shoot ratios ("ESA-truncated Walker").

### Refining areas of opportunity

The potential albedo offset map and net climate impact map cover most land areas, except Antarctica and places where our neighborhood analysis could not predict a most likely forest (e.g., core desert areas). However, some of the places with the most net negative impacts are not suitable for tree cover (e.g., the margins of deserts, the Etosha Salt Pan in Namibia). Many other locations already support tree cover or are needed for human land use. Thus, we filtered this map to places that are potentially restorable.

There are multiple maps that highlight locations potentially available to restore tree cover. We selected three that are publicly available and global in extent[2,27,31] and overlaid our net climate impact map to see how much of the identified area would result in climate-positive locations. The Griscom opportunity map was taken directly from the original publication without any additional filtering[2]. For the Bastin map, we subset their map to locations that currently had less than 25% cover but had potential for supporting greater than 25% or more tree cover[31]. For the Walker map, we included their restoration categories (i.e., R/H and R/L[27]).

Finally, we assembled spatial data from Grain for Green in China[33] and Restor[32]. From here, we quantified the pixels in which one or more projects were present using a vectorized grid snapped to the raster data. For both the Grain for Green and the Restor projects, we calculated the total value of net climate-positive (and negative) pixels in which projects were present in each 500-m pixel. We based the analysis on project presence rather than area to incorporate both datasets (GPS locations versus polygons, respectively). If there were duplicate projects in one pixel, the pixel value was only counted once in order to not over inflate the final value. Net climate-positive areas were determined as pixels with a value greater than zero, while climate-negative pixels were the inverse. The former dataset is publicly available, but we acquired the Restor data under a non-disclosure agreement that protects project privacy.

### Data availability

All input data are publicly available except the Restor data which we acquired under a non-disclosure agreement. All of the derived spatial products generated in this study have been deposited at https://doi.org/10.7910/DVN/G17RXL. This includes (1) the 24 'single land cover transition maps' (median, minimum, and maximum versions of each), (2) the most likely forest and most likely open maps (Figs. S1), (3) the potential albedo change map (Fig. S2a), as well as the minimum and maximum version, (4) the net climate impact and albedo offset map (Fig. 1), as well as an alternative net climate impact map based on a modified carbon map (Fig. S6). Source data for Figures S3 and S8 are also provided in the same data repository.

### Code availability

Code for the analyses can be found at https://doi.org/10.5281/zenodo.10672232.

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

## Acknowledgements

This work was funded by a Bezos Earth Fund grant to the Nature Conservancy and a sub-grant to Clark University from the Nature Conservancy (N.H., C.A.W., V.C.D., P.W.E., S.S., D.E.T.H., N.H.W., S.Y. and S.C.P.). Funding from the Bernina Initiative and Restor Foundation supported Restor's data collection efforts (T.W.C. and L.K.W.). We also thank the thousands of people who provided spatial data to indicate planned, current, and future projects.

## Author contributions

N.H., C.A.W., V.C.D., P.W.E., D.E.T.H., N.H.W., S.Y. and S.C.P. designed the study. NH conducted most analyses, created most of the figures, and wrote the methods. S.C.P. drafted the manuscript, and all co-authors contributed to writing and review. S.Y. conducted the analysis of actual projects. S.Y., C.A.W., and D.E.T.H. created additional figures. C.A.W. conducted the time-varying analysis. T.W.C. and L.K.W. led data collection efforts for Restor. N.H., C.A.W., V.C.D., P.W.E., S.S., D.E.T.H., N.H.W., S.Y., T.W.C., L.K.W. and S.C.P. contributed substantially to revisions.

## Competing interests

The authors declare no competing interests.
