## [Peer Review File · Nature Communications]

Accounting for albedo change to identify climate positive tree cover restorationEditorial Note: This manuscript has been previously reviewed at another journal that is not operating a transparent peer review scheme. This document only contains reviewer comments and rebuttal letters for versions considered at *Nature Communications*.

REVIEWERS' COMMENTS

Reviewer #2 (Remarks to the Author):

(Re-)review of Hasler *et al.* "Accounting for albedo change to identify climate positive tree cover restoration"

I commend the authors for their efforts invested to address the major concerns raised by myself and other two original reviewers, notably the improvement to the CO₂-eq. calculation method and the addition of a temporal dynamic sensitivity analysis. As I stated in my original review, the work is generally very thorough, and the conclusions are policy-relevant. I have only a few minor additional comments/suggestions that I think would help improve clarity surrounding some aspects of the methods as well as general flow and readability.

Abstract: "*climate-positive*" ought to be defined in the abstract.

P2, L38-40: This is an awkward description of the procedure with incorrect terminology usage. "Global warming potential (GWP)" is an emission metric with a very specific and rigid definition – i.e., the time-integrated radiative forcing of some forcing agent x relative to that of CO₂. The time horizon is completely arbitrary, and its usage here as described makes little sense.

The current phrasing of the revised Methods text on P20 L597-600 unfortunately does not add or enhance clarity here. What is the "*100-year global warming potential (GWP) of albedo relative to CO₂*"? Such a value or definition does not appear in the two references to my works (i.e., refs 25 & 26). What I think the authors have done is taken the average CO₂ fraction remaining in the atmospheric after 100 years following a unit pulse emission – computed with an IRF – and used this to normalize the output of Eq. (7). I encourage the authors to drop the "GWP" jargon and explain (around P20, L597) how (with new equation) and why the IRF was utilized, including a justification for the adopted 100-yr. time horizon.

As for the confusing text on P2 L38-40, I suggest a revision along the lines of: "*The TOA RF from albedo change is then converted to CO₂e by finding the equivalent CO₂ pulse yielding the same TOA RF when considering the average fraction of CO₂ removed from the atmosphere by all global carbon sinks after 100 years.*"

P2, L51-53: I suggest an alternate phrasing that does not insinuate that "*temperature changes*" were under scope in the study. By "*net climate outcomes*" do the authors mean specifically in the context of biogeophysical forcings? Anyway, the purpose of this sentence is to justify the decision to exclude non-albedo biogeophysical forcings for reasons stated in the previous two sentences, so perhaps it is fine to just delete it altogether to avoid the risk of confusion/misinterpretation.

P2, L67-69: Terminology like "*climate forcing*" and "*mitigation potential*" is ambiguous. I think it's clearer to use "CO₂e from albedo change" and "carbon storage potential" here, especially since these terms have already been defined or referred to previously in the manuscript.

Conclusion, P14, L367-369: *“restoring tree cover is not a substitute [...] for protection of intact forests”*. This is confusing phrasing.

Figure S7: What are the y-axis units?

Figure S9: Should these figures be area-based (i.e., missing “m⁻²” in y-axis unit labels)?

Reviewer #3 (Remarks to the Author):

Review of Hasler et al.

“Accounting for albedo to identify climate positive tree cover restoration”

Hasler et al. present a comprehensive global analysis that converts changes in surface albedo from restoring forest cover to radiative forcing equivalent of carbon dioxide (CO₂e). When restoring forest cover from open (grass, crop, etc) land cover types, albedo typically decreases and contributes to warming. When the albedo-induced warming overwhelms the cooling from carbon sequestration, the authors refer to this as a ‘net climate negative’ result. If the albedo-induced warming does not overwhelm the carbon cooling, this results in a ‘net climate positive’ effect. One of the major products of the the analysis is a global net climate radiative forcing map that indicates at 500-m spatial resolution whether a region is ‘net climate positive’ or ‘net climate negative’, presented in Figure 1.

The analysis further refines the global map using three opportunity maps: Griscom, Bastin, and Walker (Figure 3). These maps whittle down the global map to regions where the potential areas where restoration of tree cover is can be realistically implemented as a climate solution. The major finding in this step of the analysis is that there is a wide divergence – ranging from a fifth, to almost a half and to over two-thirds of areas of opportunity in the three maps listed above, respectively, that may not be suitable for restoration of tree cover as a climate solution because greater than 50% of the carbon cooling would be overwhelmed by albedo-induced warming, referred to by the authors as a ‘substantial albedo offset’.

The final step of refinement filters the opportunity maps by on-the-ground tree restoration projects around the globe. A major finding here is that less than half (45%) of the projects overlap with at least one of the three opportunity maps, indicative of a mismatch between where projects could potentially be implemented and where they actually are being implemented. Encouragingly, 84% of the project pixels are in net climate positive locations. However, nearly than a third (29%) had a substantial albedo offset (ie: greater than 50% of the albedo-induced warming offset the carbon-induced cooling), and the majority (66%) had at least a 20% offset.

The revised discussion now benefits from additional references about the social costs related to forestry and the timber market (Favero et al. 2018 and Lintunen & Rautiainen, 2021). The introduction now also notes that non-radiative biophysical global scale forcings can alter TOA fluxes and surface temperature, but methods for quantifying these effects at a global scale are not yet mature (Lawrence et al. 2022).

Overall, I find the revised manuscript to be very well written with compelling results suitable for publication in *Nature Communications*. The requested revisions from my previous review have been adequately addressed. A few minor comments are listed below:

Line 29-30: “Because tree cover often absorbs more solar radiation than other land covers, this can lead to local and—importantly for climate change

mitigation—global warming.” The placement of ‘and’ or the ‘-’ jumbles up the sentence. This should be revised.

Lines 180 &197: The use of quotations around “Griscom opportunity map” and “Walker opportunity map” seem unnecessary. Quotes are not used in Lin 188, nor are they used for Bastin. The text is cleaner without the quotes and I recommend removing them.

Lines 205-211: Figure 3. Please add *Mg Cc ha⁻¹* in small, italicized text underneath “Net Climate Impact [Mg CO₂e ha⁻¹] to match the color bar legend.

Thank you for these final suggestions! Our responses are in italics below.

Reviewer #2 (Remarks to the Author):

(Re-)review of Hasler *et al.* “Accounting for albedo change to identify climate positive tree cover restoration”

I commend the authors for their efforts invested to address the major concerns raised by myself and other two original reviewers, notably the improvement to the CO₂-eq. calculation method and the addition of a temporal dynamic sensitivity analysis. As I stated in my original review, the work is generally very thorough, and the conclusions are policy-relevant. I have only a few minor additional comments/suggestions that I think would help improve clarity surrounding some aspects of the methods as well as general flow and readability.

Abstract: *“climate-positive”* ought to be defined in the abstract.

RESPONSE: We have revised the abstract to define terms and remove jargon, also in response to the guidance from Nature Communications. We have had communications experts review the abstract for clarity for a general audience.

P2, L38-40: This is an awkward description of the procedure with incorrect terminology usage. “Global warming potential (GWP)” is an emission metric with a very specific and rigid definition – i.e., the time-integrated radiative forcing of some forcing agent x relative to that of CO₂. The time horizon is completely arbitrary, and its usage here as described makes little sense. The current phrasing of the revised Methods text on P20 L597-600 unfortunately does not add or enhance clarity here. What is the “100-year global warming potential (GWP) of albedo relative to CO₂”? Such a value or definition does not appear in the two references to my works (i.e., refs 25 & 26). What I think the authors have done is taken the average CO₂ fraction remaining in the atmosphere after 100 years following a unit pulse emission – computed with an IRF – and used this to normalize the output of Eq. (7). I encourage the authors to drop the “GWP” jargon and explain (around P20, L597) how (with new equation) and why the IRF was utilized, including a justification for the adopted 100-yr. time horizon.

As for the confusing text on P2 L38-40, I suggest a revision along the lines of: *“The TOA RF from albedo change is then converted to CO₂e by finding the equivalent CO₂ pulse yielding the same TOA RF when considering the average fraction of CO₂ removed from the atmosphere by all global carbon sinks after 100 years.”*

RESPONSE: We revised the confusing text as suggested at line 33, and edited the methods description at Lines 399-404 to match the reviewers correct account of what we performed That latter section now reads “Lastly, we normalize the CO₂e from albedo change to account for the time decay of CO₂ in the atmosphere as described by an impulse response function²⁴ for ocean and land CO₂ exchange with the atmosphere, thus dividing the output of Eq. 7 by the average CO₂ fraction remaining in the atmosphere after a representative 100 years following a unit pulse emission of CO₂”.

P2, L51-53: I suggest an alternate phrasing that does not insinuate that “*temperature changes*” were under scope in the study. By “*net climate outcomes*” do the authors mean specifically in the context of biogeophysical forcings? Anyway, the purpose of this sentence is to justify the decision to exclude non-albedo biogeophysical forcings for reasons stated in the previous two sentences, so perhaps it is fine to just delete it altogether to avoid the risk of confusion/misinterpretation.

RESPONSE: We have deleted this sentence.

P2, L67-69: Terminology like “*climate forcing*” and “*mitigation potential*” is ambiguous. I think it’s clearer to use “CO₂e from albedo change” and “carbon storage potential” here, especially since these terms have already been defined or referred to previously in the manuscript.

RESPONSE: We have deleted the term climate forcing from the paper. We retain climate mitigation potential since that is a common phrase and because we need it as a term to describe the carbon + albedo change combination.

Conclusion, P14, L367-369: “*restoring tree cover is not a substitute [...]* for protection of intact forests”. This is confusing phrasing.

RESPONSE: We have clarified this sentence. It now says “Restoring tree cover is not a panacea for climate change. It is also critical to reduce fossil fuel emissions and protect intact ecosystems.

Figure S7: What are the y-axis units?

RESPONSE: The y-axis dimensions are mass of carbon per area, and have been assigned units of kg C m⁻² with a notional maximum forest ecosystem biomass of 20 kg C m⁻². The units have now been added to the figures for each case.

Figure S9: Should these figures be area-based (i.e., missing “m⁻²” in y-axis unit labels)?

RESPONSE: Yes, great catch. The reviewer is correct, the “m⁻²” was missing from the y-axis unit labels. This has now been added.

Reviewer #3 (Remarks to the Author):

Review of Hasler et al.

“Accounting for albedo to identify climate positive tree cover restoration”
Hasler et al. present a comprehensive global analysis that converts changes in surface albedo from restoring forest cover to radiative forcing equivalent of carbon dioxide (CO₂e). When restoring forest cover from open (grass, crop, etc) land cover types, albedo typically decreases and contributes to warming. When the albedo-induced warming overwhelms the cooling from carbon sequestration, the authors refer to this as a ‘net climate negative’ result. If the albedo-induced warming does not overwhelm the carbon cooling, this results in a ‘net climate positive’ effect. One of the major products of the the analysis is a global net climate radiative forcing map that indicates at 500-m spatial resolution whether a region is ‘net climate positive’ or ‘net climate negative’, presented in Figure 1.

The analysis further refines the global map using three opportunity maps: Griscom, Bastin, and Walker (Figure 3). These maps whittle down the global map to regions where the potential areas where restoration of tree cover is can be realistically

implemented as a climate solution. The major finding in this step of the analysis is that there is a wide divergence – ranging from a fifth, to almost a half and to over two-thirds of areas of opportunity in the three maps listed above, respectively, that may not be suitable for restoration of tree cover as a climate solution because greater than 50% of the carbon cooling would be overwhelmed by albedo-induced warming, referred to by the authors as a ‘substantial albedo offset’.

The final step of refinement filters the opportunity maps by on-the-ground tree restoration projects around the globe. A major finding here is that less than half (45%) of the projects overlap with at least one of the three opportunity maps, indicative of a mismatch between where projects could potentially be implemented and where they actually are being implemented. Encouragingly, 84% of the project pixels are in net climate positive locations. However, nearly a third (29%) had a substantial albedo offset (ie: greater than 50% of the albedo-induced warming offset the carbon-induced cooling), and the majority (66%) had at least a 20% offset.

The revised discussion now benefits from additional references about the social costs related to forestry and the timber market (Favero et al. 2018 and Lintunen & Rautiainen, 2021). The introduction now also notes that non-radiative biophysical global scale forcings can alter TOA fluxes and surface temperature, but methods for quantifying these effects at a global scale are not yet mature (Lawrence et al. 2022).

Overall, I find the revised manuscript to be very well written with compelling results suitable for publication in *Nature Communications*. The requested revisions from my previous review have been adequately addressed. A few minor comments are listed below:

Line 29-30: “Because tree cover often absorbs more solar radiation than other land covers, this can lead to local and—importantly for climate change mitigation—global warming.” The placement of ‘and’ or the ‘–’ jumbles up the sentence. This should be revised.

RESPONSE: We have removed the “importantly for climate change mitigation” clause.

Lines 180 & 197: The use of quotations around “Griscom opportunity map” and “Walker opportunity map” seem unnecessary. Quotes are not used in Lin 188, nor are they used for Bastin. The text is cleaner without the quotes and I recommend removing them.

RESPONSE: We have removed the quotes.

Lines 205-211: Figure 3. Please add *Mg Ce ha-1* in small, italicized text underneath “Net Climate Impact [Mg CO₂e ha-1] to match the color bar legend.

RESPONSE: We have added this label.